# A Comprehensive Analysis of Cytokine Network in Centenarians

**DOI:** 10.3390/ijms24032719

**Published:** 2023-02-01

**Authors:** Marcello Pinti, Lara Gibellini, Domenico Lo Tartaro, Sara De Biasi, Milena Nasi, Rebecca Borella, Lucia Fidanza, Anita Neroni, Leonarda Troiano, Claudio Franceschi, Andrea Cossarizza

**Affiliations:** 1Department of Life Sciences, University of Modena and Reggio Emilia, 41125 Modena, Italy; 2Department of Medical and Surgical Sciences of Children and Adults, University of Modena and Reggio Emilia, 41124 Modena, Italy; 3Department of Surgery, Medicine, Dentistry and Morphological Sciences, University of Modena and Reggio Emilia, 41124 Modena, Italy; 4Prevention and Protection Service, University of Modena and Reggio Emilia, 41121 Modena, Italy; 5Department of Medical and Surgical Sciences, Alma Mater Studiorum University of Bologna, 40126 Bologna, Italy

**Keywords:** centenarians, cytokines, PCA, aging, immunosenescence

## Abstract

Cytokines have been investigated extensively in elderly people, with conflicting results. We performed a comprehensive analysis of the plasma levels of 62 cytokines and growth factors involved in the regulation of the immune system, in healthy centenarians, and middle-aged controls. We confirmed the previously observed increase in the levels of several pro-inflammatory cytokines, such as TNF-α and IL-6, and found that several other cytokines, directly or indirectly involved in inflammation (such as IFN-α, IL-23, CCL-5), were present at higher levels in centenarians. We did not observe any increase in the levels of anti-inflammatory cytokines, with the notable exception of the Th2-shifting cytokine IL-19. No relevant difference was observed in cytokines regulating T cell immunity. Several growth factors having a role in regulating immunity, such as G-CSF, GM-CSF, EGF, and VEGF, were upregulated in centenarians, too. Principal component analysis of the cytokine dataset showed that pro and anti-inflammatory cytokines were the variables that contributed the most to the variability of the data we observed.

## 1. Introduction

Cytokines are low molecular weight, soluble proteins that act as signal molecules and regulate an uncountable number of activities inside and outside the immune system. Cytokines are secreted by different immune cells, such as lymphocytes, macrophages or natural killer (NK) cells, and by other types of cells that interact with and modulate the function of the immune system. Changes in cytokines levels in biological fluids can provide information regarding the physiological and pathological changes of the immune system and are useful for the diagnosis and prognosis of several diseases [1].

A huge number of studies have analysed the levels of different pro- and anti-inflammatory cytokines with aging, by comparing plasma or serum levels of cytokines in young, middle aged and/or elderly subjects. In particular, an increase in the levels of pro-inflammatory cytokines is typically observed in inflammaging, a state of low-grade, subclinical chronic inflammation, which usually leads to chronic inflammatory diseases and frailty in the elderly [2]. Inflammaging is characterized by the presence of elevated levels of pro-inflammatory cytokines such as Interleukin (IL)-6, IL-1β, Tumour Necrosis Factor (TNF)-α, IL-8, IL-15, acute phase proteins, and can be identified in the elderly regardless of the degree of frailty [3,4].

However, not all aged individuals have this profile. Centenarians are considered the best example of successful ageing [5,6], as they escaped neonatal mortality, death from infectious diseases before the introduction of most vaccines and antibiotics, and fatal outcomes from age-related diseases, including cardiovascular diseases, cancer, neurodegenerative diseases [7]. Healthy centenarians and nonagenarians showed compensatory mechanisms called “immune remodeling” that control of the detrimental effects of immunosenescence [3,8,9]. In particular, they seem to cope with the chronic subclinical inflammation associated with aging through an anti-inflammatory response that balances the detrimental effects of inflammaging, therefore called “anti-inflammaging” [10].

The biological markers of healthy aging, as those observed in centenarians, are still a matter of debate [11]. Cytokines and, in particular, the “classical” pro-inflammatory cytokines, e.g., TNF-α, IL-1β and IL-6-have been extensively investigated in elderly people, with conflicting results. While there is a consensus on the age-related increase in TNF-α, and IL-6 and on their correlation with aging and inflammaging, contrasting results have been reported for other pro-inflammatory cytokines, such as IL-1β, or anti-inflammatory cytokines, including TGF-β. Increased serum levels of IL-6 is a characteristic of aging [12], which may reflect age-associated pathological processes that develop over decades, even in apparently healthy subjects [13]. IL-6 serum levels influence the onset of frailty, poor physical performance, loss of muscle strength, cognitive decline, and cardiological, neurological, and vascular events. and heart failure, cancer [14], and risk of pneumonia requiring hospitalization [15], or severe COVID-19 [16]. The picture in the case of TNF-α and IL-1β is less clear; some studies showed an involvement of IL-1β in cognitive decline and Alzheimer’s disease onset in elderly subjects [14], but others showed that IL-1β production was significantly lower in octogenarians with chronic diseases, or not associated with aging [17]. TNF-α invariably shows an age-related upregulation [18,19] and was associated with frailty, cardiovascular events and rapid cognitive decline in elderly subjects [14], but high levels of TNF-α were found to be correlated with mortality in very old subjects in one study [20], but not in another one [21].

Higher levels of anti-inflammatory cytokine IL-10 was typically observed in old subjects, but its role is debated; it was associated with a reduced risk of death from a cardiovascular event [22] and with cancer protection [23], but also with a reduced resistance to infections [24]. Finally, TGF-β levels was found inversely associated with age in one study [25], but others studies found high levels of TGF-β in elderly subjects, including centenarians [23,26].

With the aim to provide a wider picture of changes of cytokine levels with aging, we performed a comprehensive analysis of plasma cytokine levels in healthy centenarians, enrolled in the framework of previous studies, confirming the well-known increase in pro-inflammatory cytokines observed in aged people, but also showing crucial differences in the levels of chemokines and anti-inflammatory cytokines never analysed in centenarians before.

## 2. Results

We have analysed the expression of a total of 62 cytokines, chemokines, and growth factors in 22 centenarians and 34 healthy middle-aged controls. Data regarding the expression of the single cytokines are reported in the Figure 1, Figure 2, Figure 3 and Figure 4, where cytokines are grouped according to their role in the modulation of the immune response. For clarity, only cytokines and growth factors whose levels were significantly different between centenarians and controls are reported.

We first confirmed the increase in the expression of several pro-inflammatory cytokines previously observed in different cohorts of centenarians (Figure 1). In particular, we observed significantly higher levels of TNF-α, IL-6, Interferon (IFN)-α, while we could not observe a significant increase in the level of IL-1β. likely because of the sensitivity of the test used. The analysis of other pro-inflammatory cytokines also evidenced an increase in the levels of the pro-inflammatory, Th17-promoting cytokine IL-23. IL-18 levels showed a slight, despite not statistically significant, increase in centenarians. No difference has been observed in the levels of IL-12p70, of different isoforms of IL-17 and of IFN-γ.

Even if they are not considered pro-inflammatory cytokines per se, several chemokines expressed during an inflammatory response, or an infection are upregulated in centenarians. In particular, in centenarians we found higher levels of the chemokine C-X-C motif ligand (CXCL)-10, which is chemoattractant for monocytes/macrophages, T cells, NK cells, and dendritic cells. Because of a certain degree of variability among individuals, we could not detect any significant difference in the levels of other cytokines, or chemokines related to inflammation, such as CXCL-1 and CCL-2 ( Since chemokines are crucial for the orderly movement of immune system cells, we also analysed plasma levels of other chemokines which regulate either innate or adaptive immunity, and found that CCL5, CCL-19 and CX3CL-1 showed a significant increase in plasma from centenarians (Figure 3).

The increase in pro-inflammatory cytokines is often accompanied by a parallel increase in anti-inflammatory or immunomodulatory molecules. However, in our cohort we could not detect any significant change in most of the molecules with anti-inflammatory properties, such as IL-10, IL-6ra, Transforming Growth Factor (TGF)-α or TGF-β. Conversely, we observed an increase in the levels of IL-19, an anti-inflammatory, Th2-promoting cytokine.

While cytokines modulating innate, inflammatory response showed a tendency to increase with age, we could not detect relevant changes in the levels of cytokines modulating T cell-mediated, adaptive immune response. None of the cytokines regulating T cell activation or differentiation, or released by T cells after activation (namely, IL-2, IL-3, IL-4, IL-5, IL-10, IL-13, IL-17, IL-33, IFN-γ), showed significant differences between controls and centenarians. Among interleukins, only IL-11, IL-19, and IL-27 displayed a significant increase in centenarians’ plasma (Figure 2). Concerning cytokines regulating B cell response, we found an increase in the level of B-cell activating factor (BAFF), but not of A proliferation-inducing ligand (APRIL), in centenarians. The soluble form of Transmembrane activator and CAML interactor (TACI), one of the receptors of BAFF and APRIL, was also increased in these individuals (Figure 2). 

Apoptosis is a crucial process in regulating adaptive immune response, and a clear dysregulation of intrinsic and extrinsic pathways with aging have been demonstrated in several studies [27,28,29]. We found an increase in the soluble form of Fas (CD95), and of the soluble form of TRAIL, a molecule able to induce apoptosis in a caspase-8 dependent manner (Figure 3).

We finally analysed several growth factors involved in proliferation, survival and differentiation of tissues or cells undergoing dysregulation (an in particular, hypoplasia or differentiation defects) in aged people, and that can also have a role in immune system regulation (Figure 4). Both G-CSF and GM-CSF were higher in centenarians than controls. Concerning factors related to bone differentiation, BMP-2, BMP-4, BMP7, and OPN resulted upregulated in centenarians, while no difference was observed in molecules regulating adipocyte metabolism, namely Leptin and Leptin R. EGF and VEGF resulted also increased in plasma from centenarians.

To give a comprehensive picture of changes in cytokines production with age, we performed a principal component analysis of the whole data set (Figure 5).

The Principal Component (PC) 1 and 2 explain the 36% and 10.6% of the variability observed, respectively (Upper panel). Although PC2 explains 10.6% of variability of the dataset, it clearly discriminates middle-aged donors from centenarians (Panel A). In particular, pro- and anti-inflammatory cytokines are the variables that contributed the most to the PC1, while growth factors and factors related to regulation of cell survival and apoptosis contributed to PC2, further confirming that changes in the levels of cytokines involved in inflammatory regulation were the most relevant during aging (Lower panel).

Finally, as shown in Figure 6, the correlograms related to the correlations among different cytokines in young donors and in centenarians display almost the same pattern, except for BMP-2, that in centenarians displayed correlations not found in the other group.

## 3. Discussion

In this study, we performed a comprehensive analysis of cytokine levels in plasma from centenarians, compared to middle-aged controls. We confirmed most of the previous observations concerning cytokines in aging and made some new observations concerning cytokines never analysed before in centenarians. The picture that emerges is that of an increase in pro-inflammatory cytokines, which in centenarians are, at least in part, compensated by increased levels of anti-inflammatory molecules, while the analysis of cytokines related to the adaptive immune response is more complex.

We confirmed the well-established age-related increase in pro-inflammatory cytokines (and in particular of TNF-α, IL-6 and IFN-α), and we found that soluble factors with inflammatory properties, such as IL-23, were also increased in centenarians. IL-23 is a pro-inflammatory cytokine that, in a pro-inflammatory context, can be involved in differentiation of Th17 cells, especially in the presence of TGF-β and IL-6 [30]. As we did not observe any relevant variations of IL-17, the prototypical cytokine released by Th17 T cells, even in the presence of high levels of IL-23, we cannot speculate that Th17 differentiation is impaired in very old people. However, the pro-inflammatory scenario that this observation depicts is strengthened by the observed increase in other molecules that can directly or indirectly promote an inflammatory status, such as OPN, which mediates mediate apoptosis, inflammation, or vasoconstriction by using TNF-α, or Fas Ligand pathways, and GM-CSF [31]. 

As described before, an increase in anti-inflammatory molecules can often be observed in centenarians. An interesting example of this scenario is the behavior of a molecule that has never been measured, i.e., IL-19. This cytokine, belonging to the IL-10 cytokine family, is produced in response to proinflammatory stimuli [32], such as LPS, and has immunomodulatory properties; it is able to shift immune response toward a Th2 response, induces the synthesis of IL-10, and has been implicated in resolution of inflammation in neurological disorders or during neuroinflammation [32]. The high levels of IL-19 observed in centenarians could contribute to dampen a Th1-driven, inflammatory response, which could be detrimental for healthy aging. 

When we analysed the cytokines related to adaptive immune response, and in particular those regulating T cell activation and differentiation, we could not detect relevant differences between centenarians and controls. It is well known that T cell adaptive response undergoes a quantitative and qualitative decline with aging, with a reduction in the frequency of naïve T cells, the increase in terminally differentiated T lymphocytes, and a series of functional defects that results in a lower capacity to trigger de novo antigen-specific T cell responses [11]. These events are largely due to a reduced thymic output [9], and to a continuous stimulation of T cell response caused by chronic viral infections [33,34]. Our data suggest that the cytokine network regulating these events played minor, if any, role in this decline, and that no compensative mechanisms were activated to counteract this T cell dysregulation through cytokine modulation. 

In the case of B cells, centenarians showed higher levels of BAFF, but not of APRIL. BAFF and APRIL are two closely related cytokines sharing two receptors, TACI and BCMA, which are found mainly on B cells and plasma cells [35]. When B cells differentiate to plasma cells, BCMA and TACI were strongly induced [36,37]. As BAFF and APRIL are produced constitutively by monocytes, macrophages, neutrophils, among others [38], it is not surprising that we found an increase in their levels with age. It should be noted that other studies reported an inverse correlation of these cytokines with age; however, such inverse correlation has been shown in people aged 0–50 years, with primary antibody deficiency [39]. Nevertheless, soluble TACI was also increased, and this could partially reduce the effects of BAFF upregulation we observed. Indeed, the soluble form of TACI could bind BAFF in solution, so partially neutralizing the cytokine and reducing its biological activity by preventing its binding on B cells.

We previously showed that the well-known increase in Fas, both as membrane-bound protein and soluble molecule, observed with aging is compensated by a parallel downregulation of FasL in centenarians, to counteract excessive apoptosis of T cells [27]. In this cohort, we confirmed what we previously observed in the case of Fas, but not of FasL. As a non-significant trend to a reduction of FasL is present, it is likely that a larger cohort of centenarians is needed to get the statistical power needed to confirm this trend, and the observation we previously reported [27]. 

IL-11 has lymphopoietic/hematopoietic and osteotrophic properties [40], as it improves platelet recovery after thrombocytopenia and participate in the regulation of bone cell proliferation and bone resorption [41]. Thus, its increase in very old people could represent a compensatory mechanism to higher bone frailty, and to platelet reduction observed in old people. Similarly, the increase observed in several growth factors, and in particular in BMPs, likely represents an attempt to compensate defects in bone homeostasis and repair. It is interesting to note that many BMPs, besides their role in bone tissue homeostasis, play a significant role in regulating the immune system. Thus, the higher levels of BMP-4 and BMP-7 we observed, likely caused by bone frailty, could have a significant impact in determining the chronic, low level of inflammation typical of old people.

The major limitation of this study was the relatively small cohort of centenarians we were able to analyze, which reduced statistical power, and prevented any stratification. Nevertheless, even a small cohort such as this allowed us to confirm well-established phenomena, highlight changes in cytokines never investigated before, and define a global picture of cytokines changes in centenarians. Quantification of cytokine levels represents a potentially powerful tool for monitoring the immune status of patients and for adjusting therapies in several conditions, including (but not limited to) COVID-19, cancer, depression, heart disease, HIV infection, rheumatoid arthritis, or sepsis [42,43,44,45,46,47,48,49]. However, cytokines form a complex network, and it is unlikely a single cytokine provides sufficient information to predict the progression of a disease, or the success of a therapy, particularly in the case of immunomodulating therapies, or immunotherapy response [50]. As shown in this study, multiple cytokines assays provide comprehensive information about the role of immune activation and inflammation in longevity. The use of data analysis tools that provide a global evaluation of cytokines gives more information than the sum of single data, as shown in the case of cancer [51]. An artificial intelligence approach, along with the integration of cytokine data with demographic variables associated with aging and frailty will strengthen the definition and knowledge of “successful aging” and could help the diagnosis of age-related diseases and of frailty. 

In conclusion, on the one side our study confirms well-established observations concerning the upregulation of pro inflammatory cytokines in old people, on the other it adds new pieces to the cytokine mosaic of centenarians. Further analyses in other centenarians’ cohorts will help to confirm our observation, and to shed light on the complex remodeling that the immune system undergoes with aging.

## 4. Materials and Methods

### 4.1. Subjects

A total of 22 centenarians (4 M, 18 F, mean age ± SD 101.0 ± 1.2 years) and 34 control subjects (15 M, 19 F, mean age ± SD 49.0 ± 9.9 years) have been analysed. Centenarians had been previously recruited in the framework of other studies from our group. Upon consent, a structured interview was previously conducted based on a standardised questionnaire. Ten of included subjects suffered from hypertension; none of them had Type 2 Diabetes or suffered from myocardial infarction. Two subjects had and recovered from colorectal cancer. The Standardized Minimal Mental Examination Score (SMME) was 16.7 ± 4.6. Plasma was collected, immediately centrifuged twice to remove platelets and stored at −80 °C until use.

### 4.2. Cytokine Quantification

Plasma levels of 62 molecular species was quantified using a custom magnetic bead-based Luminex assay platform (Human Cytokine Discovery, R&D System, Minneapolis, MN, USA) on a a Luminex 200 Analyzer, for the simultaneous detection of the following molecules: APRIL, BAFF, BMP-2, BMP-4, BMP-7, soluble CD40L, CCL2, CCL3, CCL4, CCL5, CCL19, CCL11, CCL20, CXCL1, CXCL2, CXCL10, CX3CL1, EGF, FGF basic, FLT-3, Fas, Fas Ligand, G-CSF, GM-CSF, Granzyme B, IFN-α, IFN-β, IFN-γ, IL-1α, IL-1β, IL-2, IL-1ra, IL-3, IL-4, IL-5, IL-6, IL-6ra, IL-7, IL-10, IL-11, IL-12p70, IL-13, IL-15, IL-17, IL-17C, IL-17E, IL-18, IL-19, IL-23, IL-27, IL-33, Leptin, Leptin R, OPN, PDGF-AA, PDGF-AB/BB, PD-L1, TGF-α, TNF-α, TACI, TRAIL, VEGF. Analysis was performed according to the manufacturer’s instructions, and as previously reported [52,53,54]. Experimental data were analysed by fitting data to the standard analyte curves; measured values are the mean of two technical replicates. Data were analysed using xPONENT 3.1 (R&D System, Minneapolis, MN, USA) software.

### 4.3. Analysis of the Correlations among Plasma Cytokines in Young Donors and Centenarians

To identify the correlations among the parameters we have studied, we have designed a table containing plasma cytokines. Pairwise correlations between variables were calculated and visualized as a correlogram using R function corrplot of the R Statistical software (v4.1.2; R Core Team 2021). Spearman’s rank correlation coefficient (ρ) was indicated by color scale (orange: −1, to green: +1); significance was indicated by * *p* < 0.05, ** *p* < 0.01, and *** *p* < 0.001 inside each square. All variables were displayed using original order without applying any hierarchical clustering.

### 4.4. Statistical Analysis

Statistical analyses were performed using Prism 9.0 (GraphPad Software Inc., La Jolla, CA, USA). Student’s t test with Bonferroni’s correction has been used to compare cytokine concentrations among controls and centenarians. Principal Component Analysis (PCA) has been performed as described [55]. The datasets analysed during the current study available from the corresponding author on reasonable request.

## Figures and Tables

**Figure 1 ijms-24-02719-f001:**
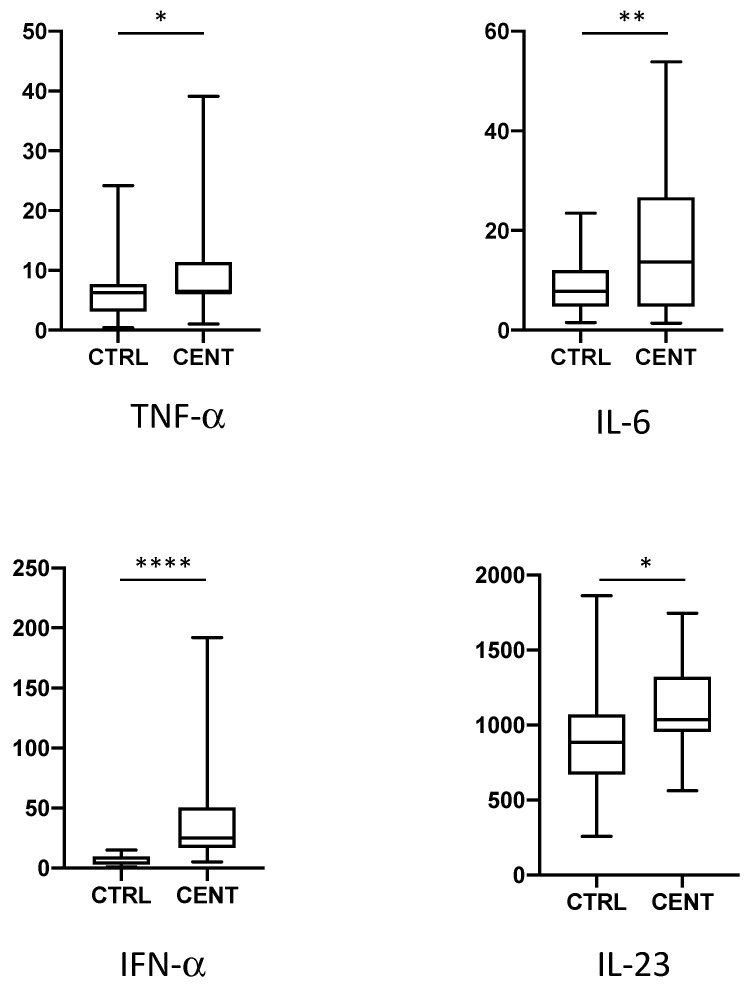
Plasma levels of pro inflammatory cytokines in controls (CTRL) and centenarians (CENT). Only cytokines whose levels were significantly different between groups are shown. Data are expressed as pg/mL. Boxes represent mean, 25 and 75 percentiles, while whiskers represent min and max values. * = *p* < 0.05; ** = *p* < 0.01; **** = *p* < 0.0001.

**Figure 2 ijms-24-02719-f002:**
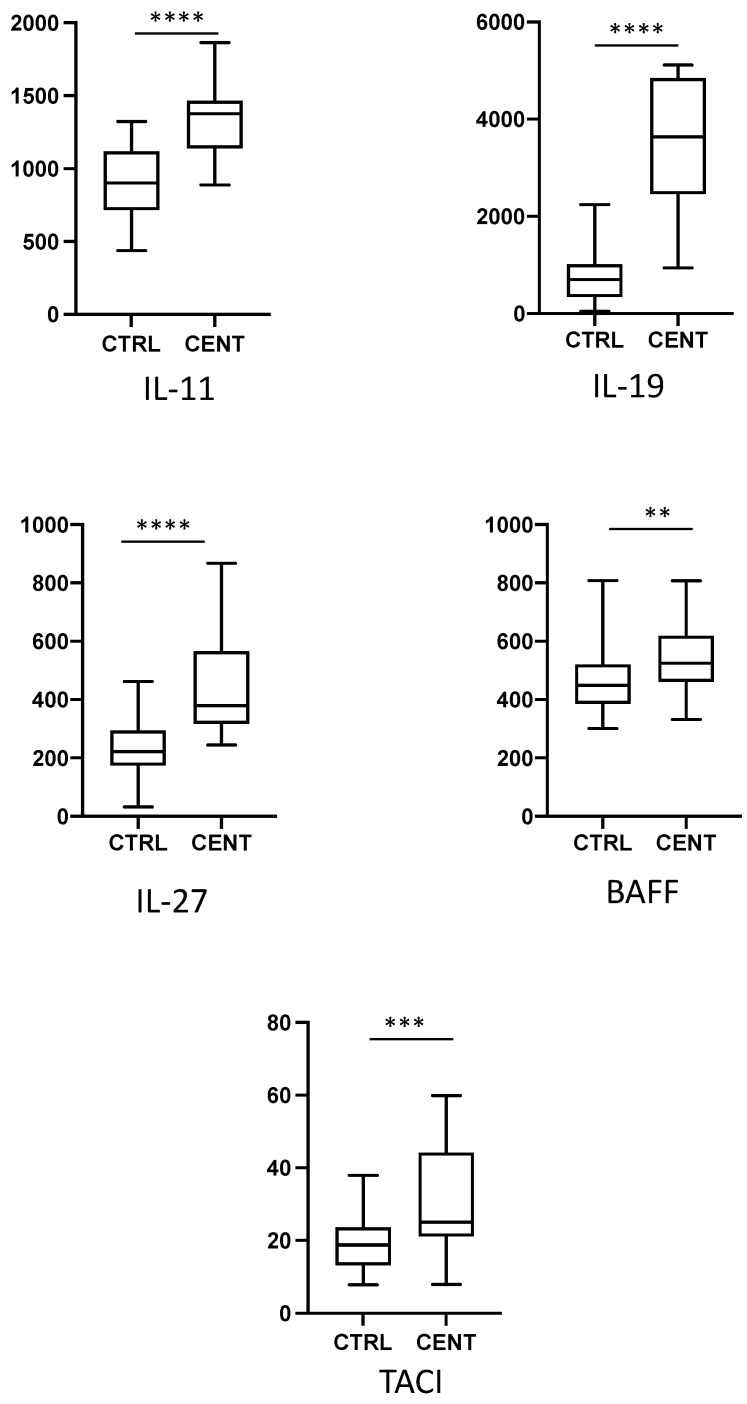
Plasma levels of cytokines regulating B and T cell differentiation in controls (CTRL) and centenarians (CENT). Only cytokines whose levels were significantly different between groups are shown. Data are expressed as pg/mL. Boxes represent mean, 25 and 75 percentiles, while whiskers represent min and max values. ** = *p* < 0.01; *** = *p* < 0.001; **** = *p* < 0.0001.

**Figure 3 ijms-24-02719-f003:**
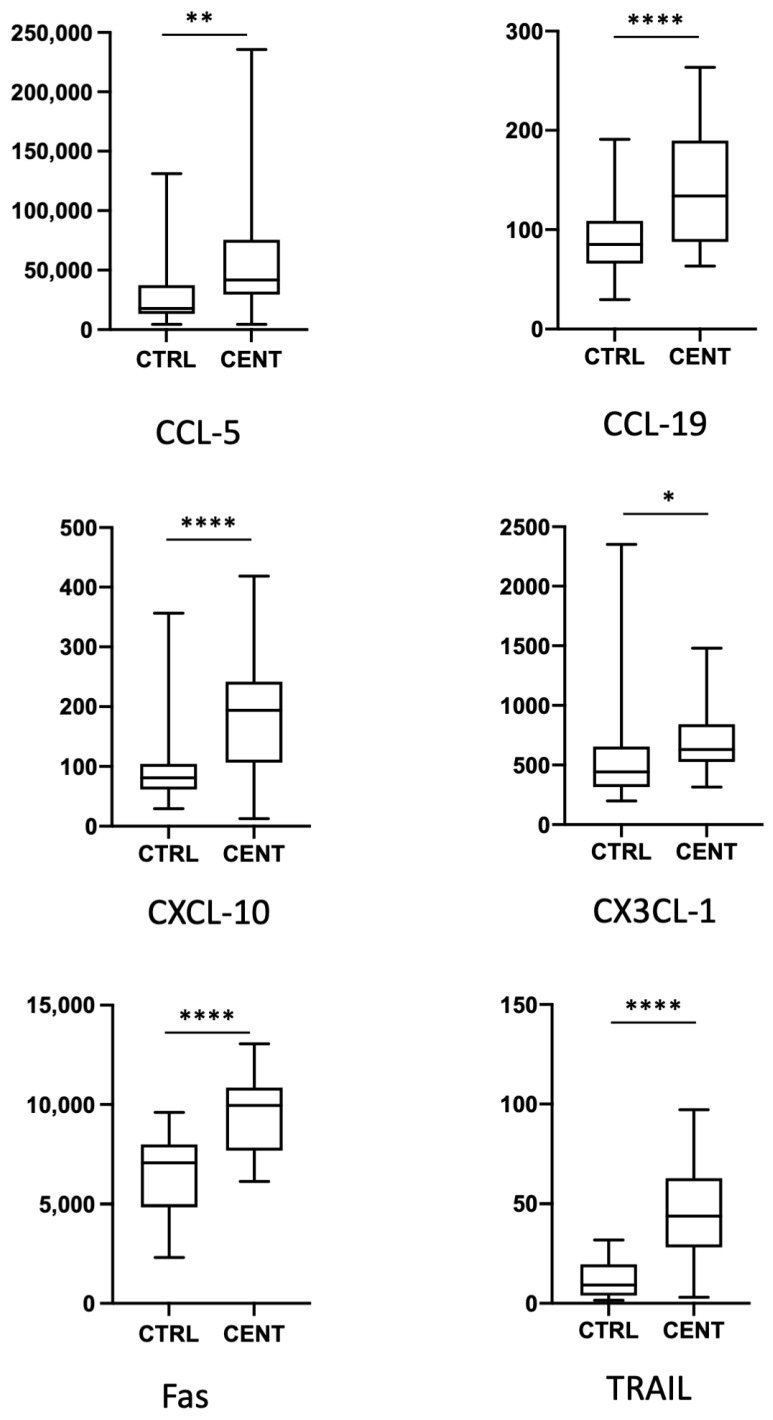
Plasma levels of chemokines and of factors regulating apoptosis in controls (CTRL) and centenarians (CENT). Only molecules whose levels were significantly different between groups are shown. Levels are expressed as pg/mL. Boxes represent mean, 25 and 75 percentiles, while whiskers represent min and max values. * = *p* < 0.05; ** = *p* < 0.01; **** = *p* < 0.0001.

**Figure 4 ijms-24-02719-f004:**
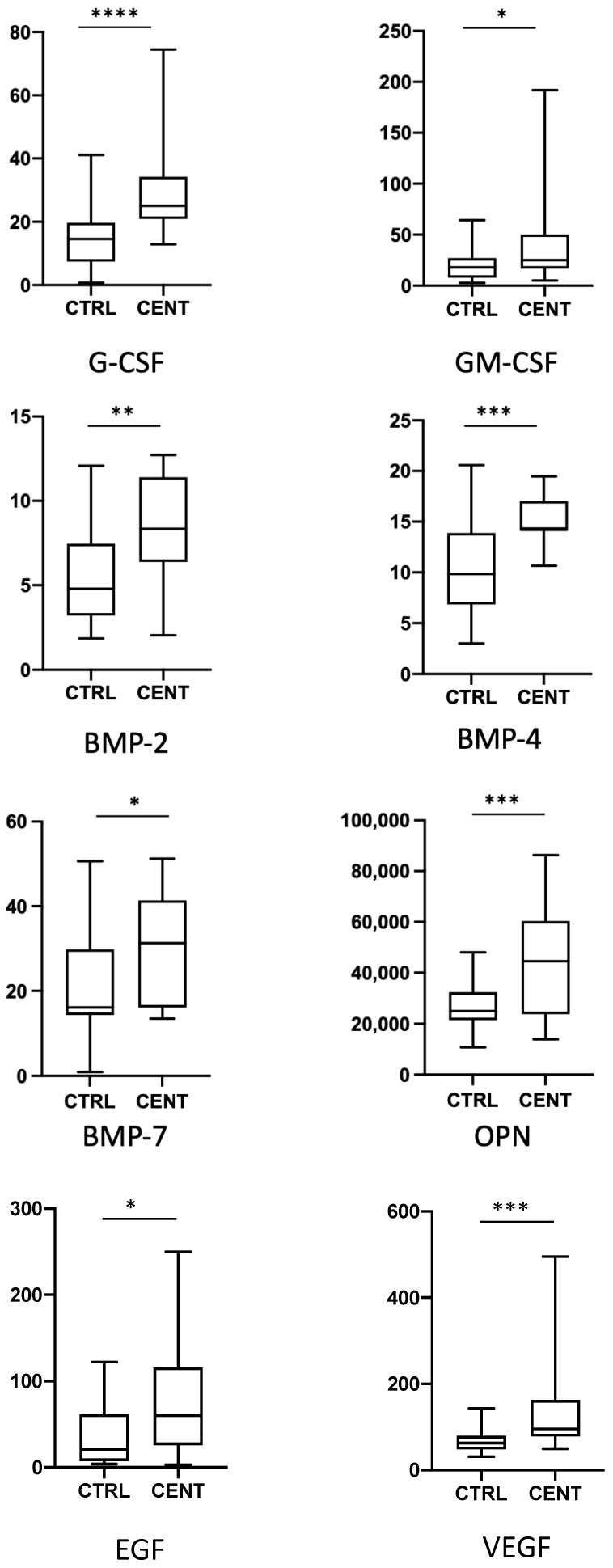
Plasma levels of G-CSF, GM-CSF, BMP-2, BMP-4, BMP-7, OPN, EGF and VEGF in controls (CTRL) and centenarians (CENT) subjects. Data are expressed as pg/mL. Boxes represent mean, 25 and 75 percentiles, while whiskers represent min and max values: * = *p* < 0.05 ** = *p* < 0.01, *** = *p* < 0.001; **** = *p* < 0.0001.

**Figure 5 ijms-24-02719-f005:**
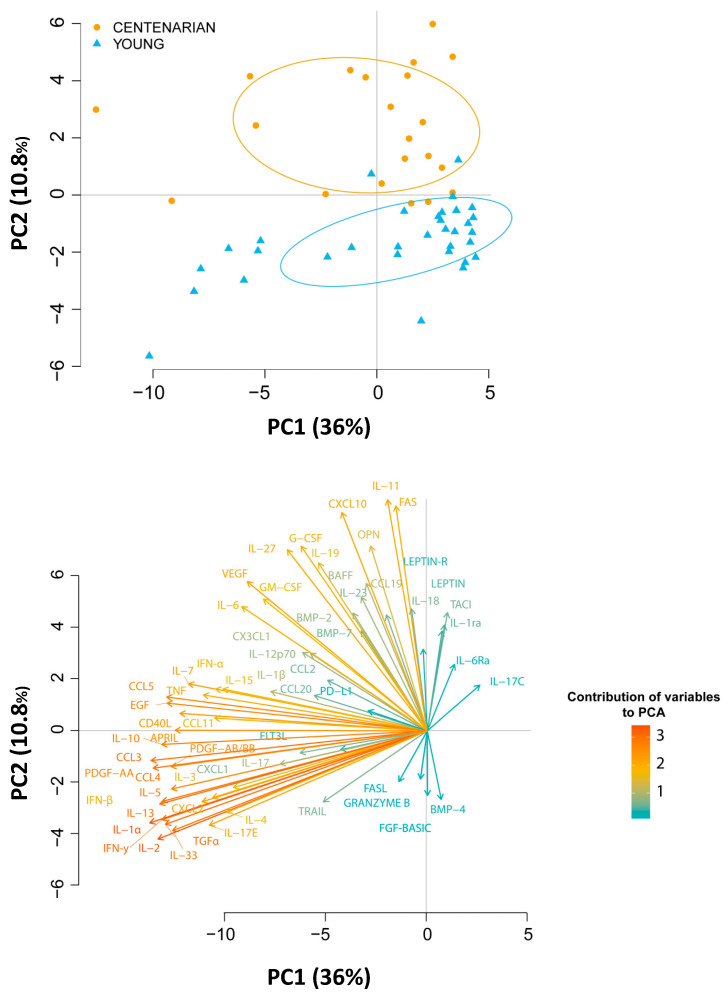
(**Upper panel**) Principal component analysis of plasma cytokine dataset in middle aged donors and centenarians. The first two principal components (PC1 and PC2) are shown. Controls are shown as, while controls are shown as light blue triangles. (**Lower panel**) contribution of the single variables to PC1 and PC2.

**Figure 6 ijms-24-02719-f006:**
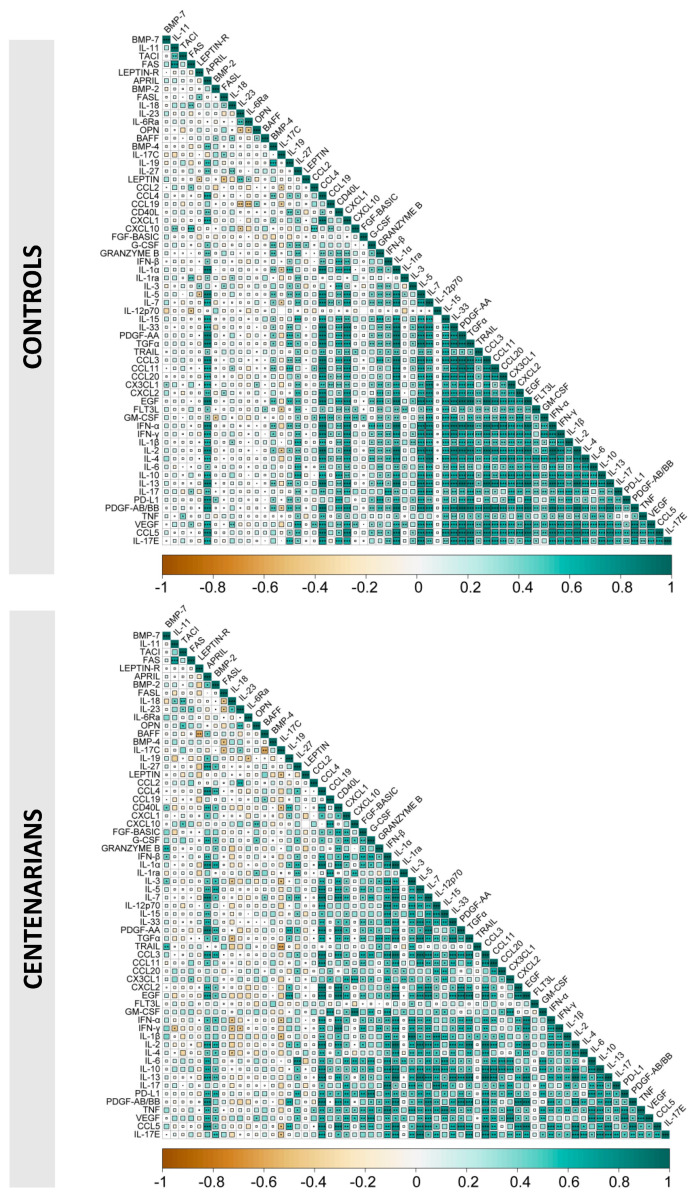
Correlations among plasma cytokines in young donors and centenarians. Correlograms show the correlations among plasma levels of the measured cytokines in control subjects (**upper panel**) and centenarians (**lower panel**). Spearman R values are shown from orange (−1.0) to green (1.0); r values are indicated by color and square size. Blank fields with dots indicate lack of signal. Statistical significance is indicated by * =*p* < 0.05, ** =*p* < 0.01, and *** =*p* < 0.001 inside each square.

## Data Availability

The datasets analysed during the current study available from the corresponding author on reasonable request.

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
