# Peer review of "A Comprehensive Analysis of Cytokine Network in Centenarians"

_ijms, 2023, doi:10.3390/ijms24032719_

Round 1

Reviewer 1 Report

The authors intended and succeeded in describing the difference of certain cytokine and growth factors involved in the regulation of the immune system in the plasma of healthy centenarians compared to middle-aged controls. They used the 62-factor Luminex platform for simultaneous detection. They confirmed the previously observed increase in the levels of several pro-inflammatory cytokines and growth factors in centenarians, such as TNF-alpha and IL-6, G-CSF, GM-CSF, EGF, and VEGF. They did not find any increase in the levels of anti-inflammatory cytokines, except the Th2-shifting cytokine IL-19, although some compensatory increase in the anti-inflammatory arsenal was expected. No relevant difference was observed in cytokines regulating T-cell immunity. 

General comments:

The article has strengths and some weaknesses. The major importance is the panel of centenarians used, who were 101 years old on average, which is a precious biological material only available to a few groups. However, there were only 22 of them, which makes statistics less reliable and prevents any stratification. Only saying »healthy centenarians « is probably not enough for more detailed discrimination of subtle differences. The conclusions should be more explicit. What is the role of cytokine measurements – is it needed or recommended in the prevention or therapy of age-related health issues and frailty? Authors could debate that.

Minor comments:

There are very few typing errors. 

Author Response

REVIEWER  1

The authors intended and succeeded in describing the difference of certain cytokine and growth factors involved in the regulation of the immune system in the plasma of healthy centenarians compared to middle-aged controls. They used the 62-factor Luminex platform for simultaneous detection. They confirmed the previously observed increase in the levels of several pro-inflammatory cytokines and growth factors in centenarians, such as TNF-alpha and IL-6, G-CSF, GM-CSF, EGF, and VEGF. They did not find any increase in the levels of anti-inflammatory cytokines, except the Th2-shifting cytokine IL-19, although some compensatory increase in the anti-inflammatory arsenal was expected. No relevant difference was observed in cytokines regulating T-cell immunity. 

General comments:

i) The article has strengths and some weaknesses. The major importance is the panel of centenarians used, who were 101 years old on average, which is a precious biological material only available to a few groups. However, there were only 22 of them, which makes statistics less reliable and prevents any stratification. Only saying »healthy centenarians « is probably not enough for more detailed discrimination of subtle differences. The conclusions should be more explicit. What is the role of cytokine measurements – is it needed or recommended in the prevention or therapy of age-related health issues and frailty? Authors could debate that.

ANSWER: We thank the reviewer for the considerations and suggestions. We fully agree with the fact that centenarians’ biological material is extremely precious, and we acknowledge that the relatively small cohort of centenarians limits the possibility to get statistical power. For this reason, we added a short sentence in the discussion, acknowledging this point as a limitation of the study:

“The major limitation of this study is the relatively small cohort of centenarians we were able to analyze, which reduces statistical power, and prevents any stratification. Nevertheless, even a small cohort like this allowed us to confirm well-established phenomena, highlight changes in cytokines never investigated before, and define a global picture of cytokines changes in centenarians.”

As suggested, we expanded the discussion in the part regarding the role of cytokine measurement in monitoring or preventing age related diseases:

“Quantification of cytokine levels represents a potentially powerful tool for monitoring the immune status of patients and for adjusting therapies in several conditions, including (but not limited to) COVID-19, cancer, depression, cardiovascular diseases, HIV infection, rheumatoid arthritis, or sepsis [39-46]. However, cytokines form a complex network, and it is unlikely a single cytokine provides sufficient information to predict the progression of a disease, or the success of a therapy, particularly in the case of immunodulating therapies, or immunotherapy response [47]. As shown in this study, multiple cytokines assays provide comprehensive information about the role of immune activation and inflammation in longevity. The use of data analysis tools that provide a global evaluation of cytokines gives more information than the sum of single data, as shown in the case of cancer [48]. An artificial intelligence approach, along with the integration of cytokine data with demographic variables associated with aging and frailty will strengthen the definition and knowledge of “successful aging” and could help the diagnosis of age-related diseases and of frailty.”

Minor comments:

i) There are very few typing errors. 

ANSWER: we carefully edited the manuscript and corrected typos.

Reviewer 2 Report

The introduction is too short, the problem should be described in more detail.

There are only abbreviations of marked parameters – the full names should be provided.

Line 19: CXCL-5 was not determined; CCL-5 was determined

Figure 1- TNF –  α is missing

In description of CXCL-10, CCL5, CCL-19 and CX3CL-1(lines 74-81) it should be added that these results are presented in Figure 3.

Lines 93-94 – IL-23 is presented in the Figure 1 not in the Figure 2

Figure 2 – results of IL-27 are presented, however this cytokine was not determined. Mayby there should be IL-17, as no results of this cytokine are presented.

In lines 90-92 the Authors inform that: „None of the cytokines regulating T cell activation or differentiation, or released by T cells after activation (namely, IL-2, IL- 91 3, IL-4, IL-5, IL-10, IL-13, IL-17, IL-17, IL-33, IFN-g) showed significant differences between controls and centenarians.” And in lines 93-94 that „Among interleukins, only IL-11, IL-19, IL-23 and IL-17 displayed a significant increase in centenarians’ plasma.” And in lines 160-161 there is; „ As we did not observe any relevant variations of IL-17,..”  As there are the opposite information regarding IL-17 - which result regarding IL-17 is true?

In most figures the first group which results are presented is a control (CTRL) and the second the centenarians (CENT). However in some figures (concerning IL-11, TACI, Fas) there is reverse order. This may mislead the reader, the order of presented groups should be the same in all figures.

Figure with BMP-7 – the abbreviations: CTRL and CENT are missing.

The results of FasL are not presented although the Authors inform (line 105) that the increase was found.

In the title of Figure 4 (line 123) the GM-CSF is missing.

As TACI is a receptor for BAFF, why Authors concluded that: „Nevertheless, soluble TACI was also increased, and this could partially reduce the effects of BAFF upregulation we observed.”(lines 196-197). It should be explained.

The information about the general health status of patients from who the blood was collected should be included.

The Authors stated that they determined 62 parameters, however in part 2.2 there are 61 abbreviations mentioned.

A brief description of the method of detection of plasma levels of cytokines should be included in part Materials and Methods.

Editorial mistakes (2x repeated the same word) lines: 92, 99, 173.

Author Response

REVIEWER 2

1)  The introduction is too short, the problem should be described in more detail.

ANSWER: We have expanded the introduction, providing more details concerning the background of the study, and the importance of cytokine production and network in the aging of the immune system, as well as some information concerning the importance of centenarians as a model of successful aging.   To support these new pieces of information, we added new references to the bibliography (Refs n. 9-23).

2) There are only abbreviations of marked parameters – the full names should be provided.

ANSWER: we have added the full name of the evaluated markers, as requested.

3) Line 19: CXCL-5 was not determined; CCL-5 was determined

ANSWER: We thank the reviewer for pointing our attention to this error, that we have promptly corrected.

4) Figure 1- TNF-α is missing

ANSWER: The reviewer is right. We added it.

5) In description of CXCL-10, CCL5, CCL-19 and CX3CL-1(lines 74-81) it should be added that these results are presented in Figure 3.

ANSWER: We added it, as requested.

6) Lines 93-94 – IL-23 is presented in the Figure 1 not in the Figure 2

ANSWER: We thank the reviewer for highlighting this error. We corrected it.

7) Figure 2 – results of IL-27 are presented, however this cytokine was not determined. Mayby there should be IL-17, as no results of this cytokine are presented.

ANSWER: Il-27 was determined, as stated in the Materials and methods section. As argued by the reviewer (see the comment below), there was a mistake in the text, where IL-17 was written instead of IL-27. Now the text is correct.

8) In lines 90-92 the Authors inform that: „None of the cytokines regulating T cell activation or differentiation, or released by T cells after activation (namely, IL-2, IL- 91 3, IL-4, IL-5, IL-10, IL-13, IL-17, IL-17, IL-33, IFN-g) showed significant differences between controls and centenarians.” And in lines 93-94 that „Among interleukins, only IL-11, IL-19, IL-23 and IL-17 displayed a significant increase in centenarians’ plasma.” And in lines 160-161 there is; „ As we did not observe any relevant variations of IL-17,..”  As there are the opposite information regarding IL-17 - which result regarding IL-17 is true?

ANSWER: We apologize for this error, and we thank the reviewer for highlighting it. IL-17 did not show any relevant change with age, and indeed the corresponding graph was not included in the figures. 

The error arose from the fact that in lines 93-94 the cytokines listed were “IL-11, IL-19, IL-23 and IL-27” (those present in figure 2) and not “IL-11, IL-19, IL-23 and IL-17”

9) In most figures the first group which results are presented is a control (CTRL) and the second the centenarians (CENT). However in some figures (concerning IL-11, TACI, Fas) there is reverse order. This may mislead the reader, the order of presented groups should be the same in all figures.

ANSWER: We thank again the reviewer for underlying this mistakes that we have promptly corrected. Those figures now show data in the same order of the others.

10) Figure with BMP-7 – the abbreviations: CTRL and CENT are missing.

ANSWER: we have added abbreviations, as requested.

11)The results of FasL are not presented although the Authors inform (line 105) that the increase was found.

ANSWER:  We checked again the data concerning FasL (reported below): as FasL did not show any significant change, we modified the sentence as follows: “We found an increase of the soluble form of Fas (CD95), and of the soluble form of TRAIL, a molecule able to induce apoptosis in a caspase-8 dependent manner (Figure 3).” 

12) In the title of Figure 4 (line 123) the GM-CSF is missing.

ANSWER: We added the “GM-CSF” in the figure 4, as requested.

13) As TACI is a receptor for BAFF, why Authors concluded that: „Nevertheless, soluble TACI was also increased, and this could partially reduce the effects of BAFF upregulation we observed.”(lines 196-197). It should be explained.

ANSWER: We thank the reviewer for giving us the opportunity to better explain this conclusion, which maybe is not obvious. As we measured the soluble form of TACI, a plausible possibility is that the soluble receptors bind the soluble BAFF, neutralizing the cytokine and reducing its biological activity by preventing its binding on the membrane form of TACI present on the B cells, the only one that can transduce a signal. A higher level of soluble TACI can reduce the effects of the higher levels of BAFF. To make this point clearer, we added the following sentence in the discussion: “Indeed, the soluble form of TACI could bind BAFF in solution, so partially neutralizing the cytokine and reducing its biological activity by preventing its binding on B cells”

14) The information about the general health status of patients from who the blood was collected should be included.

ANSWER: We added a few lines concerning the health status of subjects.

15) The Authors stated that they determined 62 parameters, however in part 2.2 there are 61 abbreviations mentioned.

ANSWER: The reviewer is right. TGF-a has been lost in the copy/paste process. It has now been restored, and is present in the list.

16) A brief description of the method of detection of plasma levels of cytokines should be included in part Materials and Methods.

ANSWER: We have briefly extended the description of the methods used for cytokines detection, by indicating the platform and software used for data acquisition; however we preferred not to add the detailed protocol, which is very standardized and straightforward.

17) Editorial mistakes (2x repeated the same word) lines: 92, 99, 173.

ANSWER: We deleted repeated words.

Round 2

Reviewer 2 Report

Page 4 – graphs in Figure 1 are repeated 2 times. The second graphs should be deleted – lines 107-8

Lines 135-136 – there is still a mess – which cytokine is presented in which Figure. IL-23 is presented in the Figure 1 while IL-11, 19, 27 are presented in the Figure 2 not 1. The Authors try to improve it but the mistakes are bigger now than in the old version.

Figure 2 – the graphs are repeated 2 times. The graphs from page No 7 should be deleted.

Figure 3 – the graphs are repeated 2 times. The graphs from page No 8 should be deleted.

Figure 4 – the graphs are repeated 2 times. The graphs from page No 10 should be deleted. There is still no GM-CSF in the title of Figure 4 (line 169) although the Authors informed that they added it.

2x repeated the same word - line 60

gramatic mistake:  „…studies showed an involvement of IL-1b is involved in cognitive decline…” - lines 71-72

a mistake in the word: immunomodulating – line 274

Author Response

Modena, 25th January 2023

Dear Editor,

I have the pleasure to send you the revised version of the manuscript entitled “A comprehensive analysis of cytokine network in centenarians” by M. Pinti et al., to be considered for publication in the International Journal of Molecular Sciences. The Reviewer #1 has no further comments, so here below we provide answers just  to the comments of the Reviewer #2.  Changes in the manuscript are highlighted by using the “Track change” function of MS Word. Here below we report the point-to-point answers to the comments of the Reviewers.

REVIEWER #2

Page 4 – graphs in Figure 1 are repeated 2 times. The second graphs should be deleted – lines 107-8

ANSWER: We thank the reviewer for the comment.  We carefully inspected the files, but we do not see any figure or graph repeated twice:  they are all present once both in the pdf and docx file. It could maybe be a visualization problem, but in the file I downloaded for revision, Figure 1 and its graphs are present once. I hope the typesetters from MDPI could solve the problem if it persists…

Lines 135-136 – there is still a mess – which cytokine is presented in which Figure. IL-23 is presented in the Figure 1 while IL-11, 19, 27 are presented in the Figure 2 not 1. The Authors try to improve it but the mistakes are bigger now than in the old version.

ANSWER: We thank the reviewer for highlighting this point: we changed “Figure 1” in “Figure 2” in line 136. Concerning IL-23, we left it in the Figure 1 as we presented this result in the context of its pro-inflammatory effect.

Figure 2 – the graphs are repeated 2 times. The graphs from page No 7 should be deleted.

ANSWER: I do not see any repetition in this figure, as in the case of Figure 1. At least in the file I downloaded, graphs of Figure 2 are present once, as well as the figure itself.

Figure 3 – the graphs are repeated 2 times. The graphs from page No 8 should be deleted.

ANSWER: As above, I do see the graphs once.

Figure 4 – the graphs are repeated 2 times. The graphs from page No 10 should be deleted. There is still no GM-CSF in the title of Figure 4 (line 169) although the Authors informed that they added it.

ANSWER: As above, I do see the figure once. Furthermore, the graph of GM-CSF levels is present in the figure 4 (top right panel of the figure).  We honestly do not know where the visualization problem comes from.

2x repeated the same word - line 60

ANSWER: we corrected it.

gramatic mistake:  „…studies showed an involvement of IL-1b is involved in cognitive decline…” - lines 71-72

ANSWER: We thank the reviewer for the comment. We corrected the grammar mistake by deleting “is involved”.

a mistake in the word: immunomodulating – line 274

ANSWER: we corrected the typo.

We thank again the reviewers for their comments, which helped us to improve the quality and clarity of our study, and we hope that now the manuscript can be considered suitable for publication in the International Journal of Molecular Sciences. 

Kind regards, 

Dr. Marcello Pinti